# Design of an evolutionary model for international trade settlement based on genetic algorithm and fuzzy neural network

Jiaqing Huang[1‡], Yang Liu[2‡], Miaomiao Tu[3]*, Osama Sohaib[4,5]

1 School of Economics, Business and Foreign Languages, Wuhan Technology and Business University, Wuhan, Hubei, China, 2 R&D department, Cabio Synthetic Biotechnology(Wuhan) Co., Ltd, Wuhan, Hubei, China, 3 School of Accounting, Wuhan Qingchuan University, Wuhan, Hubei, China, 4 School of Computer Science, Faculty of Engineering and IT, University of Technology Sydney, Ultimo, New South Wales, Australia, 5 School of Business, American University of Ras Al Khaimah, Ras Al Khaimah, United Arab Emirates

‡ JH and YL are the co-first authors of this paper.
* 13804652360@163.com

## Abstract

Accurate risk assessment in international trade settlement has become increasingly critical as global financial transactions grow in scale and complexity. This study proposes a hybrid model—Genetic Algorithm-optimized Fuzzy Neural Network (GA-FNN)—to enhance bank risk identification within this context. The objective is to improve the classification of bank-related risks by integrating the adaptability of fuzzy logic with the global optimization capability of genetic algorithms. The GA is used to fine-tune the structure, membership functions, and parameters of the FNN to improve predictive performance. Experiments were conducted on three public datasets: Bank Marketing, Lending Club, and German Credit. Results show that GA-FNN achieves an average classification accuracy of approximately 90% across high, medium, and low risk levels, outperforming traditional methods such as logistic regression, SVM (Support Vector Machine), and other metaheuristics like PSO (Particle Swarm Optimization) and SA (Simulated Algorithm). These findings demonstrate the model's effectiveness and practical value in dynamic international trade scenarios, offering a reliable approach for enhanced bank credit risk evaluation.

## 1. Introduction

In recent years, amidst the rapid unfurling of economic globalization and the resurgence of international trade post the challenges posed by the new crown, the magnitude of import and export transactions among diverse nations has witnessed a substantial augmentation. The year 2007 witnessed the eruption and swift escalation of the subprime mortgage crisis, morphing into a global financial conundrum. This upheaval led to a tumultuous oscillation in the exchange rate of the US dollar,

**Data availability statement:** All files are available from the database.(Bank Marketing Dataset: https://www.kaggle.com/datasets/janiobachmann/bank-marketing-dataset, doi: https://doi.org/10.1002/widm.1452 Lending Club: https://www.kaggle.com/datasets/wordsforthewise/lending-club,doi: 10.1109/ACCESS.2021.3079701 German Credit Risk dataset: https://www.kaggle.com/datasets/uciml/german-credit, doi: 10.24432/C5NC77)

**Funding:** The author(s) received no specific funding for this work.

**Competing interests:** The authors have declared that no competing interests exist.

exerting profound repercussions on the conventional functioning of the global economic landscape [1]. Subsequent to this, the European debt crisis emerged, instigating a credit crisis for the euro. Given the prevalent practice of utilizing both the dollar and the euro for settlement in China's import and export enterprises, the undulations in exchange rates pose a formidable impact on the economic milieu. Inherent in international trade is an inherent element of risk, permeating every facet of these transactions, with a pronounced emphasis on the susceptibility of international trade settlement junctures. The inadequately managed risks in these contexts could potentially result in the dissipation of meticulous endeavors and precipitate substantial economic losses for import and export enterprises. Hence, the discernment of risk, coupled with the financial risk assessment and credit adjudication for international trade enterprises, assumes paramount significance, constituting a pivotal imperative for the banking institutions of each nation [2].

Risk management assessment within banking has emerged as a pivotal cornerstone in the recent evolution of financial theories. Through theoretical advancements, conventional modeling methodologies have evolved into preferred tools for data analysis within numerous financial institutions. Time-honored credit risk evaluation methods, encompassing credit report scorecards, beta risk models, debt-to-income ratios, and expert judgment methodologies, have traditionally relied on historical data, financial information, and professional discernment [3]. While exhibiting a degree of interpretability, these methods encounter limitations in navigating intricate nonlinear relationships and handling extensive datasets. In contrast to these conventional approaches, machine learning methods, including logistic regression, decision trees, support vector machines, random forests, neural networks, and gradient boosters, present more adaptable modeling paradigms capable of autonomously discerning patterns and relationships within data. These methodologies excel in large-scale and high-dimensional datasets, albeit necessitating substantial training data and computational resources. Consequently, the focal point of research in machine learning for risk evaluation resides in extending data resources and refining model characteristics [4]. Machine learning methods confer substantial advantages over traditional models in the domain of credit risk assessment. Their capacity to automatically and effectively process voluminous datasets enables the precise capture of intricate nonlinear relationships, thereby enhancing the modeling efficacy in addressing real-world intricacies.

The generalization capabilities and integrated learning properties inherent in machine learning serve to augment the applicability and overall performance of the models. Moreover, the expeditious adaptability of machine learning methods to swiftly respond to evolving market conditions renders them a more supple solution for credit risk assessment [5]. Despite the commendable generalization performance of machine models such as neural networks, they may occasionally encounter challenges, including the convergence into local optima and the occurrence of gradient blow-ups during training. In contrast, meta-inspired algorithms, such as genetic algorithms and particle swarm optimization, confer distinct advantages, encompassing potent global search capabilities, resilience against local optima entrapment,

adaptability to high-dimensional spaces, high parallelism, and broad applicability across diverse disciplines in optimizing neural network models. Given the substantial uncertainties inherent in assessing the risk of international trading enterprises within the banking domain, this paper advocates a FNN-based bank risk evaluation method, offering the subsequent contributions:

1. Proposing an international trade risk assessment model based on FNN entails the comprehensive modeling of four distinct risk categories: external market risk, customer credit risk, business operation risk, and international trade financing risk.

2. The refinement of the model through Genetic Algorithm (GA) methodologies culminated in the development of the GA-FNN model, enabling high-precision identification of risks inherent in the international trade activities of banks. The identification rate across various banks consistently approached 90%.

3. Addressing the influence of the number of input features on recognition outcomes in risk assessment, the study conducted feature screening and sorting. Model testing, incorporating diverse feature dimensions across multiple banks, revealed that an augmentation in data features correlated positively with improved model performance.

In the remainder of this paper, Section 2 expounds on related work, while Section 3 establishes the GA-FNN model. Detailed accounts of experimental results and related analyses are provided in Section 4, and Section 5 is dedicated to the discussion. Conclusion is drawn at the end.

## 2. Related works

### 2.1 Bank credit risk assessment models

In recent years, bank risk management has evolved rapidly, emerging as a robust discipline with significant theoretical and practical advancements. Credit risk assessment in commercial banks, in particular, has seen deepening research and the emergence of diverse methodologies aimed at identifying and quantifying risk [6]. These methods are primarily categorized into qualitative and quantitative approaches. The qualitative analysis emphasizes non-financial indicators, relying on expert judgment, while quantitative analysis focuses on financial indicators derived from accounting data. Popular hybrid approaches include fuzzy comprehensive evaluation, which integrates both qualitative and quantitative aspects [7], entropy-based weighting methods to reduce subjectivity, and holistic evaluation models. The expert scoring method remains a widely used technique, wherein specialists evaluate a variety of risk-related factors—such as moral integrity, repayment capacity, capital adequacy, guarantees, business conditions, loan intentions, purpose, terms, collateral, and repayment strategies—to determine the overall credit risk rating [8]. The Balanced Scorecard (BSC) approach uses a standardized indicator system predefined by banks. Credit analysts assess each indicator according to the borrower's risk profile, and a weighted average score serves as the basis for determining the credit rating [9]. Model-based approaches use econometric methods to compute credit risk components, converting them into corresponding ratings. These models primarily estimate the *Probability of Default* and *Loss Given Default*. Statistical models such as the Z-score [10] and logit regression [11] rely on financial statement data, while option-theory-based models—including the KMV model, CreditMetrics, and CreditRisk+—leverage market data to assess the *Expected Default Frequency (EDF)* by analyzing the market value and volatility of a firm's assets [12,13]. These models require mature financial markets and are more suitable in well-developed financial environments.In summary, establishing precise and interpretable evaluation frameworks for credit risk necessitates the use of optimized statistical techniques and clear risk metrics, ensuring both robustness and practical applicability in financial decision-making.

### 2.2 Fuzzy neural nets and their optimisation applications

A FNN represents a computational model amalgamating a fuzzy logic system with a neural network, thereby harnessing the prowess of fuzzy systems in handling uncertainty and ambiguity alongside the capacity of neural networks to

learn and approximate intricate functions. Lin introduced an integrated fuzzy logic-decision network model, integrating self-organization and supervised learning schemes. This model evolves through a continuous optimization process, discerning fuzzy logic laws and optimal input-output affiliation functions. Extensive experiments corroborate the algorithm's superior convergence speed compared to the BP learning algorithm [14]. Buckley J, a luminary in FNN research, introduced seminal concepts like fuzzy associative memory and fuzzy cognitive maps, garnering recognition from the academic community [15]. FNN scholarship encompasses a diverse array of structures, with commonly applied forms being regularized FNNs and fuzzy associative memory neural networks [16]. Wand ingeniously combined the BP network with the Mamdani model, offering a structurally simple and conceptually transparent approach widely employed in system identification and optimization modeling [17]. Jang proposed an Adaptive Neuro-Fuzzy Inference System (ANFIS) by merging a standard five-layer neuronal network with a Takagi-Sugeno fuzzy model [18]. Traditional FNNs typically utilize the max-min rule or product rule for fuzzy inference, overlooking valuable information and incurring drawbacks. Zhang introduced a compensatory FNN by incorporating optimistic and pessimistic operations into the inference rules, addressing these limitations [19]. Given the successful application of FNNs across various domains, the growing tension between system complexity and required accuracy prompts the exploration of hybrid intelligent systems. Modern optimization methods are increasingly combined with FNNs, leveraging each other's strengths and mitigating weaknesses [20]. Through optimization and feature selection, FNNs find extensive applications in diverse fields. Murmur et al. provide a comprehensive overview of FNN models and their improved variants in applications ranging from agricultural irrigation [21]. Liu et al. utilized FNN to predict the real estate industry's development trends in the economic market [22]. Furthermore, as image processing technology advances, FNN's application in key point identification of medical images emerges as a pivotal direction [23].

The preceding research elucidates that the risk inherent in the international trade settlement process fundamentally constitutes a multivariate regression or classification problem. Traditional model methodologies, reliant on statistical calculation methods, often struggle to address outliers and achieve precise classification. The neural network approach, distinguished by its specialized activation units, markedly enhances nonlinear capabilities. In this context, FNN, leveraging fuzzy set rules, proves adept at nuanced data analysis. Consequently, this paper employs the FNN method, optimizing its pertinent performance, to realize bank risk assessment within the international trade resolution process.

## 3. Methodology

### 3.1 Genetic algorithms

To position the use of Genetic Algorithms (GA) within a broader context, it is important to consider recent developments in hybrid intelligent systems for credit risk assessment. Metaheuristic algorithms such as Particle Swarm Optimization (PSO), Differential Evolution (DE), and Simulated Annealing (SA) have gained attention for their effectiveness in optimizing complex, nonlinear models. While each technique offers unique advantages, GA remains widely used due to its robustness, global search capability. GA stands as an optimization algorithm emulating natural selection and hereditary mechanisms, tailored for addressing search and optimization challenges. Anchored in Darwin's evolutionary theory, GA progresses through generations, simulating biological inheritance, mutation, selection, and crossover mechanisms to converge toward the optimal solution. The fundamental steps of the algorithm encompass population initialization, fitness assessment, selection, crossover, mutation, and the generation of a new population [24]. In the initialization phase, a population is randomly generated, with each individual representing a potential solution to the problem. Fig 1 illustrates the parent generation and its subsequent mutation process.

Following the completion of crossover and variation, the fitness assessment phase scrutinizes the results by evaluating the fitness value for each individual's solution to the problem. This phase gauges the quality of solutions by calculating their fitness values. In the selection phase, individuals are chosen based on their fitness values, often employing methods

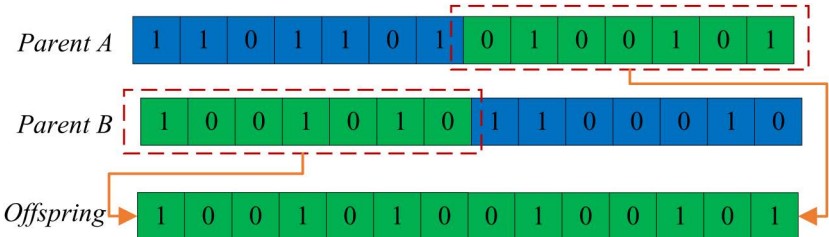

### Binary genes Crossover

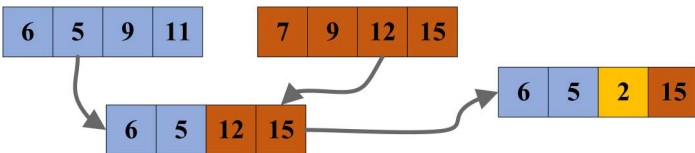

### Crossover and mutation

**Fig 1. GA crossover and mutation process.**

like roulette selection, where individuals with higher fitness have a greater likelihood of being selected. The crossover phase engenders a new individual by simulating a genetic crossover operation that amalgamates genetic information from two individuals. The mutation phase introduces random mutations, altering the genetic information of individuals to expand the search space [25].

The process involves generating new populations through operations such as crossover, selection, and mutation, iterating these steps until stopping conditions are met. The fitness function, denoted as f(x), corresponds to the objective function for the individual in the search space, and the conversion method is expressed as Fit(f(x)).

$$\text{Fit}\,(f(x)) = \begin{cases} f(x) & \text{maximum the problem} \\ -f(x) & \text{minimize the problem} \end{cases} \tag{1}$$

Although fitness function is simple and intuitive, it can not satisfy the requirement of non-negative probability in the roulette wheel luxury selection, and some of the functions to be solved are very different from each other in the distribution of function values. For the problem of finding the minimum value, do the following transformation.

$$\text{Fit}\,(f(x)) = \begin{cases} c_{max} - f(x) & f(x) < c_{max} \\ 0 & \text{others} \end{cases} \tag{2}$$

where $c_{max}$ is an appropriate relatively large number, is the $f(x)$ is the maximum estimate of the value. For the problem of finding the maximum value, do the following transformation.

$$\text{Fit}\,(f(x)) = \begin{cases} c_{min} + f(x) & f(x) > c_{min} \\ 0 & \text{others} \end{cases} \tag{3}$$

where $c_{min}$ is a suitable relatively small number, is the $f(x)$ the minimum estimate of the value. If the objective function is a minimisation problem, then,

$$\text{Fit}(f(x)) = \begin{cases} \frac{1}{1+c+f(x)} & c \geq 0, c + f(x) \geq 0 \end{cases}$$

(4)

If the objective function is a maximum problem, then

$$\text{Fit }(f(x)) = \begin{cases} \frac{1}{1+c-f(x)} & c \geq 0, c - f(x) \geq 0 \end{cases}$$

(5)

where, the $c$ is a conservative estimate of the bounds of the objective function. The iteration of the model progresses by calculating the fitness function, thereby generating the relevant operators and ultimately achieving the optimization of the objective problem. To align more closely with the objectives of this study, it is important to clarify the specific parameters used in the Genetic Algorithm. In this context, GA is primarily employed to optimize the structure and weights of the neural network, where the optimization objective is to minimize the prediction error—typically measured by mean squared error—and to enhance classification accuracy. Key parameters include population size, crossover rate, mutation rate, and the number of generations. By fine-tuning these parameters, GA enables a global search of the solution space, which helps avoid local minima—a common limitation of traditional gradient-based methods. This targeted optimization directly contributes to improved predictive performance in credit risk assessment, enhancing the reliability of risk classification outcomes in banking scenarios.

### 3.2 Fuzzy neural networks

FNN is a computational model that amalgamates fuzzy logic and neural networks, with the goal of addressing problems characterized by uncertainty and ambiguity. FNNs leverage the fuzzy reasoning capability of fuzzy systems, coupled with the learning and adaptive abilities inherent in neural networks. They find wide applications in control systems, pattern recognition, and decision support.

The typical structure of FNNs comprises fuzzy inference units and neural network units [26]. The fuzzy inference unit employs fuzzy set theory to handle uncertainty, reasoning through a set of fuzzy rules. On the other hand, the neural network unit focuses on learning and adaptive tuning, utilizing algorithms such as backpropagation to optimize weight parameters. This optimization process ensures that the model adeptly adjusts to inputs and generates the desired output. Fuzzy logic neurons, a prevalent class of fuzzy neurons in FNNs, exhibit the input-output relationship of:

$$u = I(x, w)$$ (6)

$$y = f(u - \theta)$$ (7)

where $x = (x_1, x_2, \ldots, x_N)$ is the neuron input, and $\omega = (\omega_1, \omega_2, \ldots, \omega_N)$ is the neuron weight, and $u$ is the neuron state, and $y$ represents the neuron output, and $\theta$ is the neuron threshold, and $f$ is the output function, and $I$ is the fuzzy logic function.

The concept of the affiliation function is introduced in fuzzy mathematics, providing a means to express the degree of affiliation of each element to a fuzzy set. This function serves as an intuitive representation of fuzziness and vagueness, facilitating the quantification of fuzzy sets. The accurate establishment of the affiliation function is pivotal for appropriately expressing fuzzy sets, serving as the foundation for employing precise mathematical methods to analyze and process fuzzy information. Common affiliation functions include the Gaussian function, as depicted in equation (8), and the sigmoid function, as illustrated in equation (9):

$$\mu(x) = \text{gaussian}(x; c, \sigma) = \exp\left(-\left(\frac{x-c}{\sigma}\right)^2\right)$$

(8)

$$\mu(x) = \frac{1}{1 + exp(-a(x-c))}$$

(9)

After completing the design of the fuzziness of the neural network parameters we established a FNN model, the structure of which is presented in Fig 2:

In Fig 2, the FNN is delineated into five layers: the input layer, the fuzzification layer, the fuzzy inference layer, the normalization layer, and the output layer. Each node in encapsulates specific fuzzy rules. The fuzzified data undergoes matching with the fuzzy rules in the fuzzy inference layer, concurrently assessing the fitness of each rule. The fitness values are determined based on preceding fuzzy inference rules. The output layer, often referred to as the defuzzification layer, finalizes the crisp output of the FNN.

$$y_i = \sum_{j=1}^{m} \omega_{ij} \bar{a}_j, i = 1, 2, \ldots, r$$

(10)

Where, the $y_i$ is the result through the output layer. Through the introduction of the FNN network model above, it can be seen that the learning parameters of the FNN include the value of the affiliation function and its $c_{ij}$ value and its $\sigma_{ij}$ value; the other is the value of the last output layer of the FNN. Although the structural principles of FNNs are presented, it is necessary to detail how specific network parameters are defined and utilized in the risk evaluation setting. For example, parameters such as the number of fuzzy rules, membership function types, and the architecture of the rule layer and output layer should be specified. In this study, the FNN is tailored to map both qualitative (e.g., management quality) and quantitative (e.g., debt ratios) inputs into a comprehensive risk score. The performance objective is to improve classification consistency and interpretability, particularly in borderline cases of creditworthiness. This structure allows for the modeling of ambiguity and subjectivity inherent in financial risk factors, which traditional networks often struggle to capture effectively.

### 3.3 GA-FNN model based on GA optimization

After completing the description and analysis of the FNN, we have further illustrated the proposed GA-FNN model based on GA optimization, the overall flow of which is shown in Fig 3:

In the constructed model, corresponding to the common types of risks in international trade, data is input into the model based on four columns of indicators. The specific types and formats of data will be elaborated in the subsequent section. This data is then fed into the model for training, aiming to obtain the identification of high, medium, and low-risk categories.

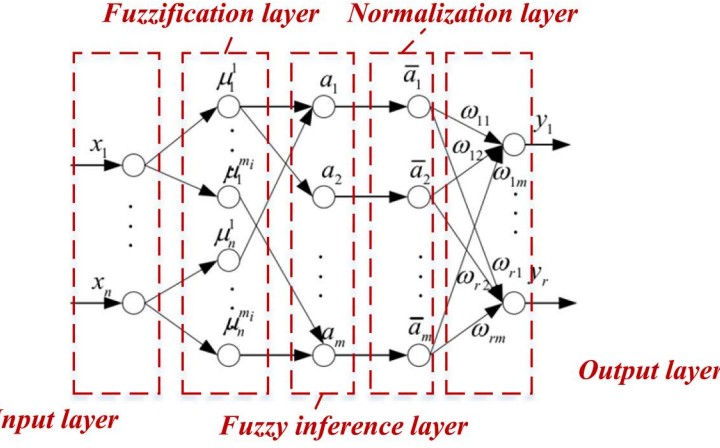

**Fig 2. The structure for the FNN.**

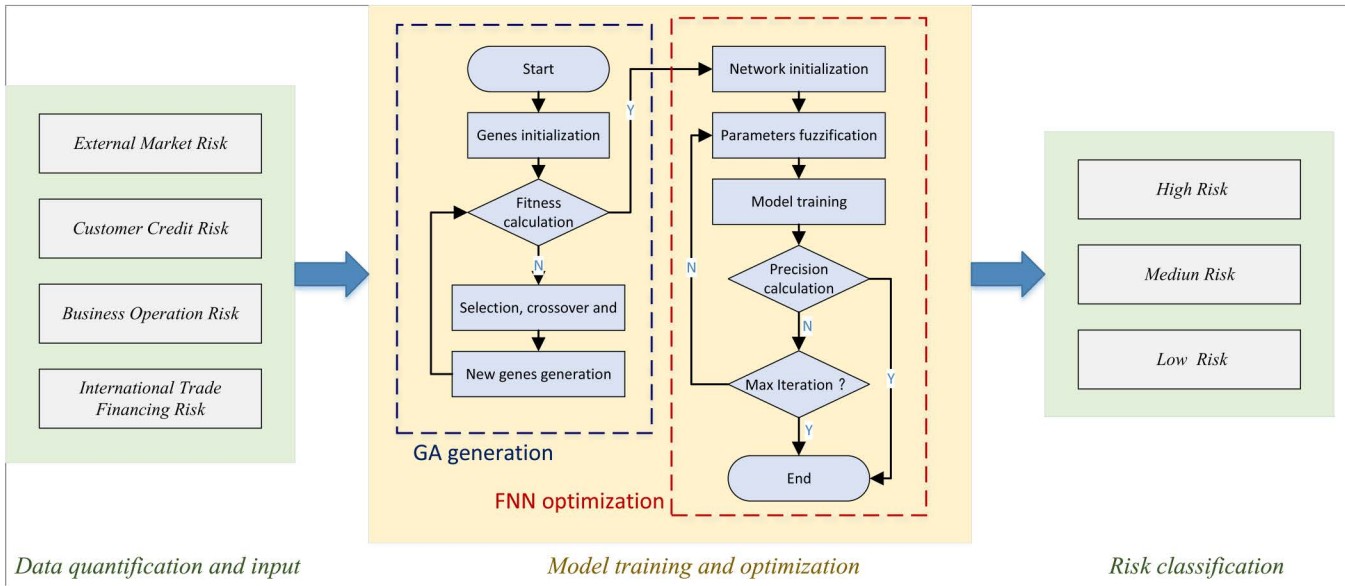

**Fig 3. The framework for the proposed model and GA-FNN.**

In the model training and optimization module depicted in Fig 3, the process initiates by generating a population. Subsequently, fitness results are calculated based on the generated population, and an assessment is made to determine if the set requirements are met. This assessment guides the decision to proceed to the next step of network initialization or to refine the generated population data for optimization. Following the optimization of population data, these refined data are utilized to initialize the network model. Parameters are then initialized according to the specifications of the network construction process. Subsequently, the model undergoes standard training procedures. During model training, both training and test data are proportionally fed into the model, with the aim of optimizing the characteristics of these particles through theGA algorithm to achieve model optimization. The conclusion of training is marked by judgments based on accuracy and maximum training, respectively. The integration of GA with FNN aims to exploit the strengths of both methods by using GA to optimize FNN parameters such as membership functions, fuzzy rule weights, and network topology. The specific optimization objective is to minimize classification error on a credit risk dataset while maximizing generalization performance on unseen data. This hybrid model addresses both the structural learning of the FNN and its parameter tuning, ensuring that the resulting model is not only accurate but also adaptable to varying data distributions. In the context of bank risk analysis, this means more robust prediction of borrower default risk under diverse economic conditions. This section thus serves as a critical link between algorithmic design and its applied relevance, bridging the gap between methodological development and financial practice.

## 4. Experiment result and analysis

### 4.1 Experiment setup

For the analysis, public data from three banks were selected: the Bank Marketing Dataset [27], Lending Club [28], and the German Credit Risk dataset [29]. The data encompass primary indicators related to solvency, operating capacity, profitability, and development capacity, with a focus on ordinary credit. Considering the characteristics of international trade and generalizing the indicators, this paper categorizes the risks of the three banks in international trade into four categories: external market risk, customer credit risk, business operation risk, and international trade financing risk. These categories

are subjected to quantitative analysis, with each level of indicators analyzed and quantified through their common three secondary indicators. This results in a total of 12 dimensional characteristics being analyzed. Public datasets from Kaggle and official sources—namely the Bank Marketing, Lending Club, and German Credit datasets—were used in this study. Prior to modeling, all datasets underwent standard data cleaning procedures. Missing values were handled using mean imputation for numerical features and mode imputation for categorical features. Categorical variables were encoded using one-hot encoding, and numerical variables were standardized to zero mean and unit variance. Feature engineering included combining related attributes and removing low-variance features to improve model efficiency and stability.

The comparison methods employed include the traditional model RAROC [30], LR, BPNN, SVM, and FNN. The RAROC risk model serves as an effective evaluation model for assessing the business risk of enterprises and banking business risk. It aids in the quantitative evaluation of banks through RAROC indicators, offering insights for capital structure adjustment, rational fund allocation, and decision-making basis for business risk evaluation, interest rate setting, and risk control in international trade financing. The traditional model RAROC is widely used in traditional risk control processes, providing a basis for data analysis through non-neural network methods. In the evaluation index, risks are categorized as high, medium, and low, transforming the problem into a multi-classification problem. Consequently, this paper focuses on assessing the accuracy of the model's evaluation. For the RAROC model, expected loss (EL), economic capital (EC), and cost of capital were calculated using standard Basel formulas, with $EL = PD \times LGD \times EAD$, and EC based on a 99.9% confidence level. The cost of capital was set at 10% to reflect typical industry standards. For the Logistic Regression (LR) model, we performed grid search with 5-fold cross-validation to tune the regularization type (L1 vs. L2) and penalty strength ($C \in \{0.01, 0.1, 1, 10\}$). The best results were achieved with L2 regularization and $C = 1$. Features were standardized to improve model convergence.

To ensure transparency and reproducibility, we clarify several implementation details of the GA-FNN model. The fuzzy rule base was constructed based on expert knowledge and data distribution, resulting in 12 fuzzy rules covering the major risk categories. Triangular membership functions were used for simplicity and computational efficiency. On the GA side, the initial population was randomly generated with a size of 50. A crossover rate of 0.8 and a mutation rate of 0.1 were selected based on preliminary tuning. The stopping criterion was set as either reaching 100 generations or achieving no improvement in the last 10 iterations. Regarding model evaluation, this study focuses on classification accuracy as the primary metric. Although we acknowledge the value of other metrics such as F1-score, sensitivity, and specificity, the current dataset presents relatively balanced class distributions, and our primary goal is to assess the overall identification capability of the model rather than individual class-specific performance. Furthermore, due to limited data availability, the dataset was split into training (70%), validation (15%), and testing (15%) subsets without cross-validation. We recognize that future work should incorporate more robust evaluation methods and explore the model's stability under volatile financial conditions.

## 4.2 Risk recognition and model evaluation

After completing data cleaning and training the relevant model, we proceeded with the analysis of the model's recognition accuracy. Initially, we conducted risk identification on the Bank Marketing Dataset, achieving the categorization of high, medium, and low risks. Subsequently, we compared the effects of various methods, including RAROC, LR, BPNN, SVM, FNN, and others. Before delving into algorithm comparisons, we conducted a data analysis during the optimization process of the Genetic Algorithm (GA) methods. We analyzed the number of optimization iterations post-model training and optimization. The results for different numbers of iterations are illustrated in Fig 4.

From Fig 4, it is evident that as the number of model iterations increases, the average recognition rate of GA-FNN exhibits continuous improvement, eventually reaching a relatively stable state. Following the optimization by Genetic Algorithm (GA), method comparison experiments were conducted. The RAROC risk model serves as a scientific evaluation model for effectively assessing enterprise business risk and banking business risk. Utilizing RAROC indicators, banks

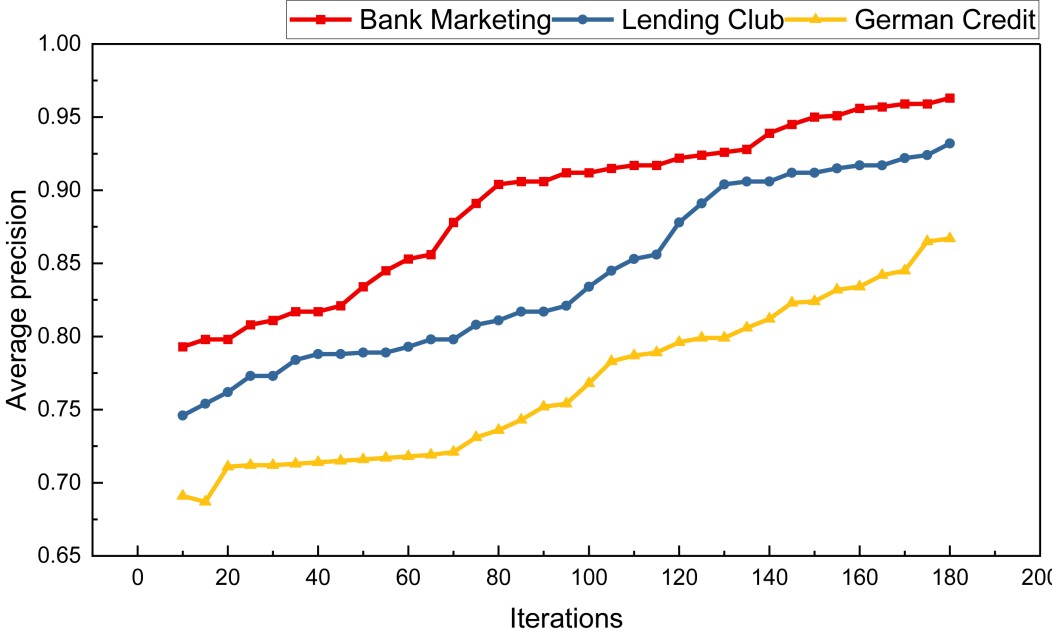

**Fig 4. The GA iteration result.**

can quantitatively assess their position, aiding in strengthening capital structure adjustment, rational fund allocation, and facilitating decision-making based on business risk evaluation, interest rate development, and risk control management in international trade financing. The RAROC model primarily focuses on revenue and cost risk analysis, providing a foundation for decision-making.

In addition to the traditional RAROC model, we also employed classic machine learning methods for risk analysis. The identification results are depicted in Fig 5.

From Fig 5, it is evident that, for the Bank Marketing Dataset, the overall recognition rate is consistently above 90%. This high recognition rate is attributed to the completeness of the bank's data and the clarity of the international trade process, ensuring a robust recognition performance.

After completing the analysis of the Bank Marketing Dataset, we extended our data analysis to include the Lending Club and the German Credit Risk Dataset. The results of these analyses are presented in Figs 6 and 7.

In Figs 6 and 7, it is apparent that after Genetic Algorithm (GA) optimization, the recognition capability of the FNN model has seen notable improvement. This improvement is evident in both the Lending Club and German Credit Risk datasets, with the recognition rate exceeding 90% in both datasets. The recognition accuracy, as indicated by the recognition rate, is marginally lower for Lending Club and German Credit Risk compared to the Bank Marketing Dataset. This could be attributed to the inherent complexity of the bank's own data, contributing to a slight reduction in the recognition rate.

To further validate the model performance and assess the impact of optimization methods on performance enhancement, we conducted tests using different optimization methods. The results are depicted in Fig 8.

Two additional commonly used optimization algorithms, Simulated Annealing (SA) and Particle Swarm Optimization (PSO), were selected for model optimization. As observed in Fig 8, the overall performance improves when optimized using different algorithms. However, the degree of enhancement is not as pronounced as when utilizing the Genetic Algorithm (GA). Nevertheless, this analysis underscores the significance of employing meta-heuristic algorithms for updating recognition results.

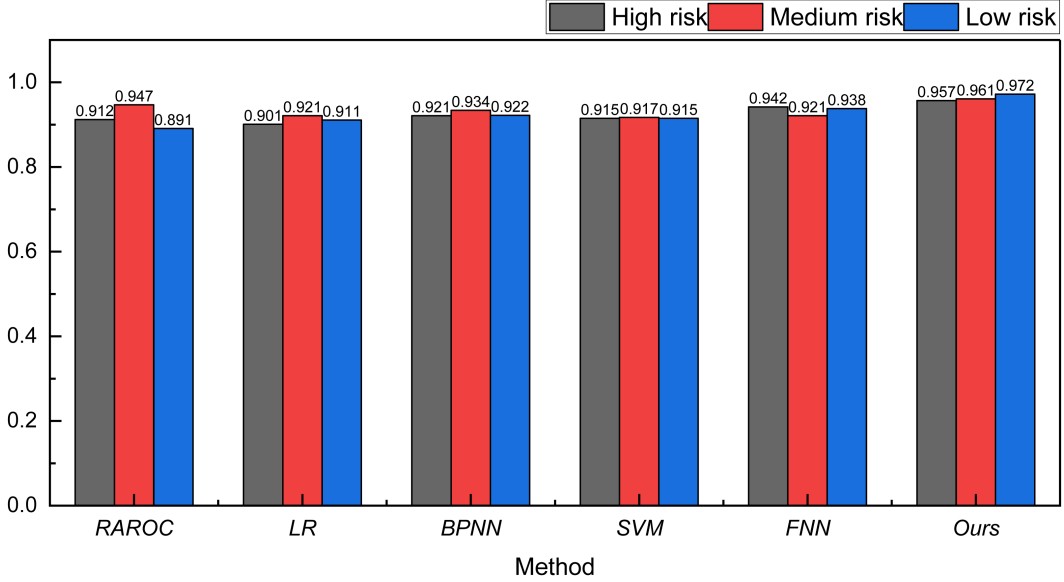

**Fig 5. The result for the risk assessment on Bank Marketing Dataset bank.**

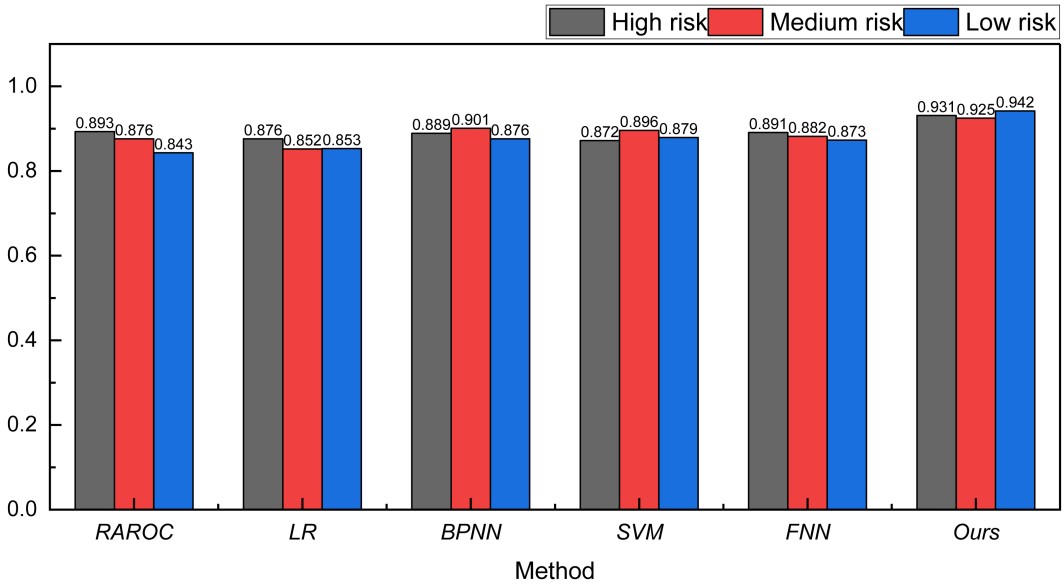

**Fig 6. The result for the risk assessment on lending club.**

Simultaneously, we computed the average classification results of different optimization methods across various datasets. The average recognition rates for high, medium, and low risks are presented in Table 1:

As evident from Table 1, the disparity in average recognition results across different optimization methods is minimal. However, the Genetic Algorithm (GA) method, despite its simplicity, outperforms the other two methods in terms of overall risk recognition rate. Specifically, under the Bank Marketing Dataset, the three-category recognition rate for GA reaches 96.3%, surpassing SA's 92.6% and PSO's 94.9%.

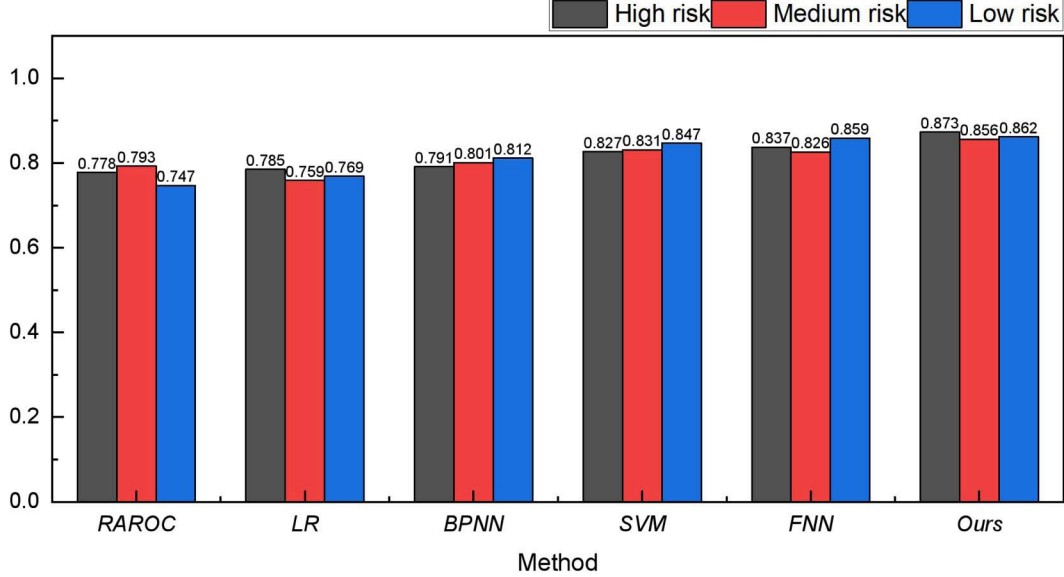

**Fig 7. The result for the risk assessment on German credit risk.**

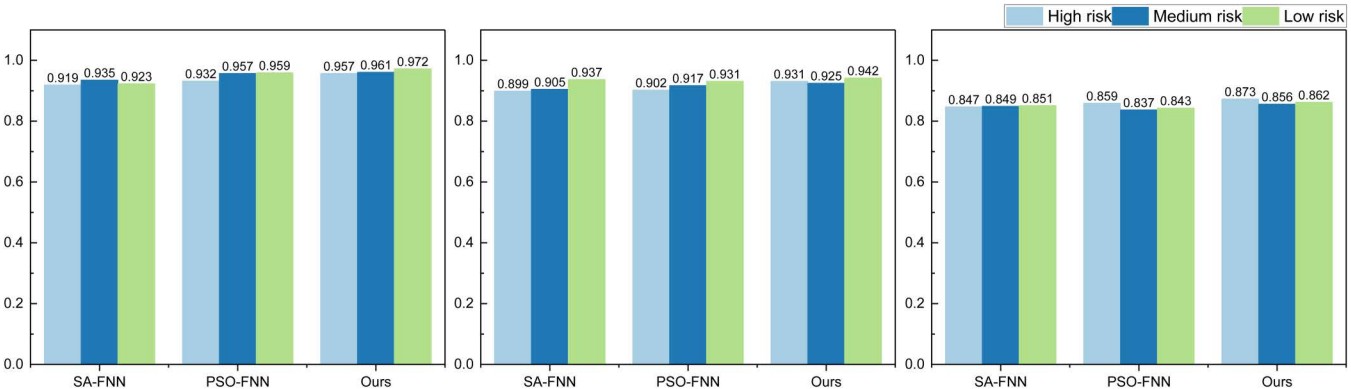

**Fig 8. The result for the risk assessment on three datasets using different optimal methods.**

**Table 1. The average risk recognition result using different optimisation methods.**

| Methods | Bank Marketing Dataset | Lending Club | German Credit Risk |
|---|---|---|---|
| SA-FNN | 0.926 | 0.914 | 0.849 |
| PSO-FNN | 0.949 | 0.917 | 0.846 |
| Ours | 0.963 | 0.933 | 0.864 |

### 4.3 Feature selection and analysis

In international trade, the issue of missing data is prevalent, potentially causing challenges in risk prediction. Therefore, it becomes essential to test the identification results of the model under different features. We selected different dimensional features to analyze the results based on the characteristics of the data, and the outcomes are illustrated in Fig 9.

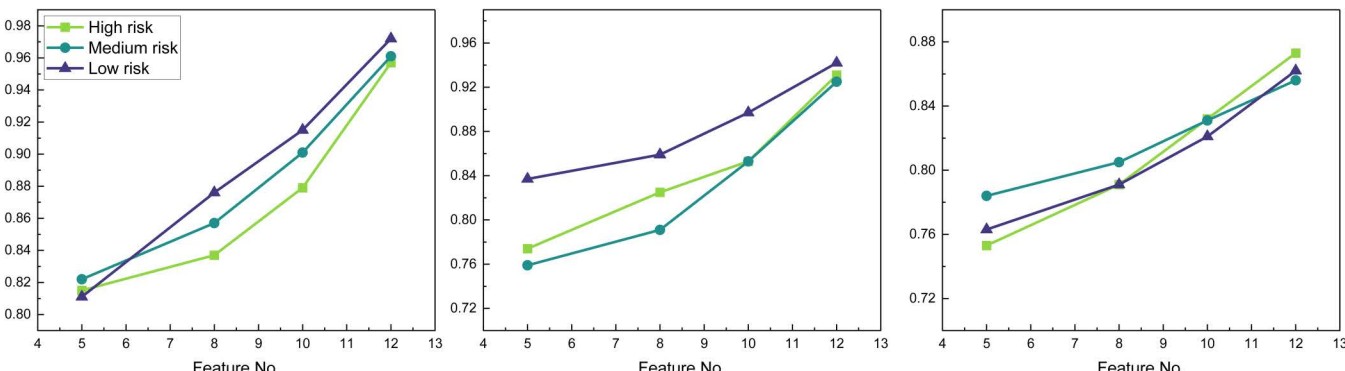

**Fig 9. The result for the risk assessment on A, B, C bank using different features.**

In the results presented in Fig 9, we utilized the proposed GA-FNN to test different feature dimensions, and the three graphs in the figure depict the results of data testing in three banks A, B, C. From the figure, it is evident that with the increase of feature dimensions, the overall recognition effect is enhanced to a certain extent, reaching optimal efficiency when using all features. The comparison of different features implies that in the realm of international trade financial risk prediction, model performance can be significantly improved by the continuous increase of data features.

## 5. Discussion

This study investigated bank risk assessment in the context of international trade settlement using a hybrid FNN optimized via a GA, benchmarking it against conventional statistical and machine learning models. Beyond methodological comparison, our analysis focused on how different datasets affect model performance and interpretability. The datasets used—Bank Marketing [27], Lending Club [28], and the German Credit Risk dataset—differ significantly in data structure, feature richness, and class distribution, which directly influenced the model's behavior. For example, the German Credit dataset has a limited number of features and is relatively balanced, allowing the FNN model to converge quickly with high accuracy. In contrast, the Lending Club dataset contains high-dimensional and imbalanced loan data, which challenged general classifiers such as LR and SVM, yet highlighted the strength of FNN in handling fuzzy boundaries between risk categories. Variable influence analysis revealed that features such as "loan purpose," "employment length," and "default history" had varying degrees of impact on the final classification depending on the dataset. In the Lending Club dataset, borrower intent and historical performance played a dominant role, whereas in the German dataset, credit history and savings level were more influential. The inclusion of fuzzy membership functions allowed FNN to capture these nonlinear, context-dependent relationships more effectively than crisp models.

Regarding model optimization, a comparative test between GA, PSO, and SA showed that while PSO exhibited faster convergence in early iterations, GA achieved more stable and higher-quality solutions over repeated runs. Despite GA's conceptual simplicity, its performance in refining rule weights and membership parameters within the FNN framework proved effective for our credit risk classification task. While our evaluation primarily relied on accuracy, this decision was driven by class balance in the datasets and the goal of evaluating holistic model performance. Nonetheless, we acknowledge that incorporating additional metrics such as F1-score and sensitivity would provide more nuanced insight— especially in datasets with class imbalance like Lending Club. Future research should expand the indicator set to include macroeconomic variables and behavioral features, which could further improve prediction under volatile market conditions. With the rapid growth of international economic and trade collaboration, the characteristics of international trade and settlement methods are becoming increasingly diversified and conspicuous. The close connection with international trade financing has emerged as a significant trend in the development of commercial banks' international business, serving as a

crucial indicator to gauge the modernization and internationalization levels of these banks. It is essential to acknowledge that international trade financing risks are becoming more intricate, and effectively managing these risks is a crucial element in commercial bank risk management. While banks can mitigate risks to some extent through the automated identification of risks using artificial intelligence methods, our model training revealed that risk identification shows a certain degree of improvement with the advancement of model optimization and feature performance. Therefore, fully considering the impact and interconnection of international trade financing and settlement business, it is imperative to construct a system of assessment indicators. This system should be based on relevant financial indicators of the enterprise, comprehensive development indicators of the enterprise, and additional risk assessment indicators reflecting the actual business information of international trade financing products. These include categories of international trade financing products, financing periods, financing amounts, default costs, international trade settlement modes, and more. Constructing a system of assessment indicators aligned with the risk characteristics of international trade financing is the key to addressing such challenges.

## 6. Conclusion

This paper proposed a GA-FNN model for bank risk identification in the context of international trade settlement, integrating fuzzy neural networks with genetic algorithm optimization. The model quantified four major categories of risk—external market, customer credit, operational, and trade financing—based on 12 representative input indicators. Through empirical evaluation on three public datasets, the GA-FNN consistently outperformed baseline methods, including Logistic Regression, SVM, and traditional BPNN, with an improvement of approximately 2%–5% in accuracy across datasets. The method achieved average classification accuracies of 0.963, 0.933, and 0.863 in identifying high-, medium-, and low-risk categories, respectively. Comparative analysis also demonstrated that GA produced more stable and effective optimization results than PSO and SA under the same experimental conditions. These findings validate the robustness and adaptability of the GA-FNN framework in bank credit risk assessment, offering practical implications for improving risk control in international trade finance scenarios.

In our research, we observed that the model's performance improvement is more pronounced with different feature data. In future studies, we aim to enhance the generalization performance of the current model by expanding the range of applicable risk indicators. Additionally, optimizing feature dimensions and identifying optimal features through screening will be areas for further research.

## Acknowledgments

We thank the anonymous reviewers whose comments and suggestions helped to improve the manuscript.

## Author contributions

**Conceptualization:** Jiaqing Huang, Osama Sohaib.

**Data curation:** Jiaqing Huang, Yang Liu, Miaomiao Tu, Osama Sohaib.

**Investigation:** Yang Liu.

**Methodology:** Yang Liu.

**Resources:** Osama Sohaib.

**Validation:** Jiaqing Huang, Miaomiao Tu.

**Visualization:** Osama Sohaib.

**Writing – original draft:** Jiaqing Huang, Miaomiao Tu.

**Writing – review & editing:** Yang Liu, Miaomiao Tu.

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
