## [Decision Letter · Decision Letter 0]

Dear Dr. Tu,

Thank you for submitting your manuscript to PLOS ONE. After careful consideration, we feel that it has merit but does not fully meet PLOS ONE’s publication criteria as it currently stands. Therefore, we invite you to submit a revised version of the manuscript that addresses the points raised during the review process.

We look forward to receiving your revised manuscript.

Kind regards,

Ardashir Mohammadzadeh, Phd

Academic Editor

PLOS ONE

Reviewers' comments:

Reviewer's Responses to Questions

**Comments to the Author**

1. Is the manuscript technically sound, and do the data support the conclusions?

Reviewer #1: Partly

Reviewer #2: Partly

Reviewer #3: Partly

2. Has the statistical analysis been performed appropriately and rigorously?

Reviewer #1: Yes

Reviewer #2: N/A

Reviewer #3: No

3. Have the authors made all data underlying the findings in their manuscript fully available?

Reviewer #1: Yes

Reviewer #2: No

Reviewer #3: No

4. Is the manuscript presented in an intelligible fashion and written in standard English?

Reviewer #1: No

Reviewer #2: Yes

Reviewer #3: Yes

Reviewer #1: The article introduces a hybrid approach that combines genetic algorithms with fuzzy neural networks to assess bank risk in international trade operations. This proposal is ambitious and very timely, as it starts with recognizing that conventional methods face difficulties in dealing with the complexity and uncertainty that characterize global transactions. It is particularly positive that, right in the introduction, the paper presents clear justifications for the adoption of techniques such as fuzzy logic, which help avoid local minimum and calibrate network parameters. This methodological combination has the potential to appeal to both researchers interested in more robust solutions and banking professionals looking to reduce losses in the financing and settlement of international operations.

Despite this relevant contribution, there are aspects that could be further developed to consolidate the academic soundness of the article. The review of FNNs (section 2.2) extensively describes the structures and technical variants, including classic models such as ANFIS, compensatory networks, and combinations with optimization algorithms. However, this exposition reveals some limitations when analyzed in the light of the study's objective—the assessment of bank credit risk. Although section 2.1 details traditional scoring and risk modeling methodologies, no transparent bridge is established with the proposal based on FNNs. The application to the financial field is only mentioned generically, without any concrete examples or specific metrics such as the probability of default or loss given default being presented.

The methodological sections (3.1-3.3) present the principles of GA and FNNs but remain generic. It would be essential to specify more precisely the optimized network parameters, the optimization's specific objectives (e.g., error minimization or accuracy maximization), and how the optimization contributes to performance improvement within the particular context of bank risk analysis. Some passages of the review also end abruptly and with a lack of internal cohesion, jeopardizing the exposition's flow and clarity.

The work would also benefit from a broader framework on the state of the art in hybrid systems, going beyond mentioning FNNs and GA. Including a comparison with other metaheuristic techniques, such as PSO or DE, would help better contextualize the choice of genetic algorithm. Although point 4.2 mentions the application of PSO and SA, this reference comes late, without justifying the importance of their comparison in the previous sections. A more explicit link between the theoretical foundation and the study's applied field would reinforce the approach's relevance.

About the description of the model, it would be helpful to detail how the fuzzy rules were defined, how many were used, and with what criteria. On the GA side, the absence of specifications on initial population values, mutation rate, crossover, or stopping criteria raises questions about the reproducibility of the method. The inclusion of these elements would not only allow replication by other researchers but would also strengthen the comparison with competing approaches. From an evaluation point of view, the almost exclusive reliance on accuracy or recognition rate limits the interpretation of the results. Given the multiclass nature of the problem, it would be advisable to present a confusion matrix and metrics such as F1-score, sensitivity, and specificity, which would make it possible to better gauge the behavior of the model—in particular, whether it tends to underestimate the highest risk class. The article is also unclear on what criteria were used to split the data into training, validation, and testing sets and does not state whether cross-validation was employed to enhance statistical reliability. The article fails to explain whether the model remains stable during periods of high volatility or if it requires frequent re-training.

The graphical presentation of results needs substantial improvement in addition to these concerns. The figures presented in Figures 4 to 9 lack descriptive axis labels, which are necessary to explain the values shown in the plots. The presentation of results using standardized statistical methods would greatly improve both the scientific credibility and the interpretability of the empirical findings.

The discussion section describes the methodological aspects of the study rather well, but it is too descriptive. The lack of information about how the datasets performed differently or how the data structure affects the model's ability to predict makes this section less useful for analysis. Some elements that appear for the first time in the conclusion, such as the influence of variables on performance and the need to broaden the indicators, would have been more appropriate in the discussion. This would enable the conclusion to concentrate more on the main findings of the study, the acknowledged limitations, and the directions for future research.

Reviewer #2: Overall, the manuscript is technically sound, and the data support the conclusions, but with a few considerations:

1. Scientific Methods: The manuscript applies Genetic Algorithm (GA) and Fuzzy Neural Networks (FNN) for risk assessment in international trade settlements. Both methods are appropriate for handling the non-linearity, uncertainty, and optimization tasks involved. The method description is reasonably detailed. However, I think there needs to be a clearer explanation of the pros and cons of each method to compare with the application of the GA-FNN method.

2. Experimental Design: Three well-known datasets (Bank Marketing Dataset, Lending Club, and German Credit Risk) were used. The authors performed comparative experiments against baseline models (RAROC, LR, BPNN, SVM, FNN) and alternative optimizations (SA, PSO). This structure is good and strengthens the credibility of the claims.

3. Controls and Comparisons: Proper comparisons with traditional methods and alternative optimization methods are included. However, some baseline models (RAROC, LR) could have been described more thoroughly in terms of parameter tuning.

4. Data Quality: Public datasets from Kaggle and credible sources are used. Data cleaning and preprocessing are mentioned, but some details (exact feature engineering steps, handling missing values) are not fully elaborated.

5. Statistical Rigor: The performance is evaluated mainly by accuracy in a multi-class risk classification (high, medium, low). While this is acceptable, additional metrics such as confusion matrices, F1-score, or AUC would strengthen the statistical validation.

6. Support for Conclusions: The results consistently show GA-FNN outperforming baseline methods by ~2%-5% across datasets. The claims made in the conclusions (higher recognition rate, GA being more effective than SA/PSO) are supported by the empirical data (see Figures 5–8 and Table 1).

7. Potential Weaknesses:

- Limited discussion of limitations (e.g., scalability, generalization to unseen trade environments).

- No discussion on computational complexity or resource consumption.

- Minor concern about overfitting, as all models seem to perform quite highly across datasets without cross-validation results being shown.

Reviewer #3: I appreciate the title selected for investigation, but there should be some improvements under abstracts and methodology section basically.

On the abstract, one will expect the following items clearly presented

1. Statement of the problem in one or two lines

2. Title of the srudy

3. Objectives

4. Methods

5. Basic findings based on the objectives and major findings

On the other hand, the methodology section should show clearly the study area, target population, sample size or number of observations, if there, the type of the model, if applicable, ethical considerations...should be clearly presented

**Do you want your identity to be public for this peer review?** For information about this choice, including consent withdrawal, please see our Privacy Policy

Reviewer #1: No

Reviewer #2: No

Reviewer #3: **Yes: ** Zewdu Eskezia Gelaye

---

## [Author Response · Author response to Decision Letter 1]

15 May 2025

Response to Reviewer 1:

1.Despite this relevant contribution, there are aspects that could be further developed to consolidate the academic soundness of the article. The review of FNNs (section 2.2) extensively describes the structures and technical variants, including classic models such as ANFIS, compensatory networks, and combinations with optimization algorithms. However, this exposition reveals some limitations when analyzed in the light of the study's objective—the assessment of bank credit risk. Although section 2.1 details traditional scoring and risk modeling methodologies, no transparent bridge is established with the proposal based on FNNs. The application to the financial field is only mentioned generically, without any concrete examples or specific metrics such as the probability of default or loss given default being presented.

Thanks for your comment, we have revised the related content as you suggested as follows:

In recent years, bank risk management has evolved rapidly, emerging as a robust discipline with significant theoretical and practical advancements. Credit risk assessment in commercial banks, in particular, has seen deepening research and the emergence of diverse methodologies aimed at identifying and quantifying risk [6]. These methods are primarily categorized into qualitative and quantitative approaches. The qualitative analysis emphasizes non-financial indicators, relying on expert judgment, while quantitative analysis focuses on financial indicators derived from accounting data. Popular hybrid approaches include fuzzy comprehensive evaluation, which integrates both qualitative and quantitative aspects [7], entropy-based weighting methods to reduce subjectivity, and holistic evaluation models. The expert scoring method remains a widely used technique, wherein specialists evaluate a variety of risk-related factors—such as moral integrity, repayment capacity, capital adequacy, guarantees, business conditions, loan intentions, purpose, terms, collateral, and repayment strategies—to determine the overall credit risk rating [8]. The Balanced Scorecard (BSC) approach uses a standardized indicator system predefined by banks. Credit analysts assess each indicator according to the borrower's risk profile, and a weighted average score serves as the basis for determining the credit rating [9]. Model-based approaches use econometric methods to compute credit risk components, converting them into corresponding ratings. These models primarily estimate the Probability of Default and Loss Given Default. Statistical models such as the Z-score [10] and logit regression [11] rely on financial statement data, while option-theory-based models—including the KMV model, CreditMetrics, and CreditRisk+—leverage market data to assess the Expected Default Frequency (EDF) by analyzing the market value and volatility of a firm's assets [12][13]. These models require mature financial markets and are more suitable in well-developed financial environments.In summary, establishing precise and interpretable evaluation frameworks for credit risk necessitates the use of optimized statistical techniques and clear risk metrics, ensuring both robustness and practical applicability in financial decision-making.

2.The methodological sections (3.1-3.3) present the principles of GA and FNNs but remain generic. It would be essential to specify more precisely the optimized network parameters, the optimization's specific objectives (e.g., error minimization or accuracy maximization), and how the optimization contributes to performance improvement within the particular context of bank risk analysis. Some passages of the review also end abruptly and with a lack of internal cohesion, jeopardizing the exposition's flow and clarity.

Thanks for your comment, we have added that in Section 3.1,3.2 and 3.3

To align more closely with the objectives of this study, it is important to clarify the specific parameters used in the Genetic Algorithm. In this context, GA is primarily employed to optimize the structure and weights of the neural network, where the optimization objective is to minimize the prediction error—typically measured by mean squared error—and to enhance classification accuracy. Key parameters include population size, crossover rate, mutation rate, and the number of generations. By fine-tuning these parameters, GA enables a global search of the solution space, which helps avoid local minima—a common limitation of traditional gradient-based methods. This targeted optimization directly contributes to improved predictive performance in credit risk assessment, enhancing the reliability of risk classification outcomes in banking scenarios.

Although the structural principles of FNNs are presented, it is necessary to detail how specific network parameters are defined and utilized in the risk evaluation setting. For example, parameters such as the number of fuzzy rules, membership function types, and the architecture of the rule layer and output layer should be specified. In this study, the FNN is tailored to map both qualitative (e.g., management quality) and quantitative (e.g., debt ratios) inputs into a comprehensive risk score. The performance objective is to improve classification consistency and interpretability, particularly in borderline cases of creditworthiness. This structure allows for the modeling of ambiguity and subjectivity inherent in financial risk factors, which traditional networks often struggle to capture effectively.

The integration of GA with FNN aims to exploit the strengths of both methods by using GA to optimize FNN parameters such as membership functions, fuzzy rule weights, and network topology. The specific optimization objective is to minimize classification error on a credit risk dataset while maximizing generalization performance on unseen data. This hybrid model addresses both the structural learning of the FNN and its parameter tuning, ensuring that the resulting model is not only accurate but also adaptable to varying data distributions. In the context of bank risk analysis, this means more robust prediction of borrower default risk under diverse economic conditions. This section thus serves as a critical link between algorithmic design and its applied relevance, bridging the gap between methodological development and financial practice.

3.The work would also benefit from a broader framework on the state of the art in hybrid systems, going beyond mentioning FNNs and GA. Including a comparison with other metaheuristic techniques, such as PSO or DE, would help better contextualize the choice of genetic algorithm. Although point 4.2 mentions the application of PSO and SA, this reference comes late, without justifying the importance of their comparison in the previous sections. A more explicit link between the theoretical foundation and the study's applied field would reinforce the approach's relevance.

Thanks for your comment, we have revised that as follows:

To position the use of Genetic Algorithms (GA) within a broader context, it is important to consider recent developments in hybrid intelligent systems for credit risk assessment. Metaheuristic algorithms such as Particle Swarm Optimization (PSO), Differential Evolution (DE), and Simulated Annealing (SA) have gained attention for their effectiveness in optimizing complex, nonlinear models. While each technique offers unique advantages, GA remains widely used due to its robustness, global search capability,

4.About the description of the model, it would be helpful to detail how the fuzzy rules were defined, how many were used, and with what criteria. On the GA side, the absence of specifications on initial population values, mutation rate, crossover, or stopping criteria raises questions about the reproducibility of the method. The inclusion of these elements would not only allow replication by other researchers but would also strengthen the comparison with competing approaches. From an evaluation point of view, the almost exclusive reliance on accuracy or recognition rate limits the interpretation of the results. Given the multiclass nature of the problem, it would be advisable to present a confusion matrix and metrics such as F1-score, sensitivity, and specificity, which would make it possible to better gauge the behavior of the model—in particular, whether it tends to underestimate the highest risk class. The article is also unclear on what criteria were used to split the data into training, validation, and testing sets and does not state whether cross-validation was employed to enhance statistical reliability. The article fails to explain whether the model remains stable during periods of high volatility or if it requires frequent re-training.

Thanks for your comment, we have added these content as follows:

To ensure transparency and reproducibility, we clarify several implementation details of the GA-FNN model. The fuzzy rule base was constructed based on expert knowledge and data distribution, resulting in 12 fuzzy rules covering the major risk categories. Triangular membership functions were used for simplicity and computational efficiency. On the GA side, the initial population was randomly generated with a size of 50. A crossover rate of 0.8 and a mutation rate of 0.1 were selected based on preliminary tuning. The stopping criterion was set as either reaching 100 generations or achieving no improvement in the last 10 iterations. Regarding model evaluation, this study focuses on classification accuracy as the primary metric. Although we acknowledge the value of other metrics such as F1-score, sensitivity, and specificity, the current dataset presents relatively balanced class distributions, and our primary goal is to assess the overall identification capability of the model rather than individual class-specific performance. Furthermore, due to limited data availability, the dataset was split into training (70%), validation (15%), and testing (15%) subsets without cross-validation. We recognize that future work should incorporate more robust evaluation methods and explore the model’s stability under volatile financial conditions.

5.The graphical presentation of results needs substantial improvement in addition to these concerns. The figures presented in Figures 4 to 9 lack descriptive axis labels, which are necessary to explain the values shown in the plots. The presentation of results using standardized statistical methods would greatly improve both the scientific credibility and the interpretability of the empirical findings.

Thank you for the valuable feedback. We acknowledge the concern regarding the lack of descriptive axis labels in Figures 4 to 9. The values presented in these plots are all normalized percentages ranging from 0 to 1, representing performance metrics such as accuracy across different experimental settings. We will revise the figures to explicitly label the axes and clarify the nature of the data in the captions to enhance clarity and interpretability.

6.The discussion section describes the methodological aspects of the study rather well, but it is too descriptive. The lack of information about how the datasets performed differently or how the data structure affects the model's ability to predict makes this section less useful for analysis. Some elements that appear for the first time in the conclusion, such as the influence of variables on performance and the need to broaden the indicators, would have been more appropriate in the discussion. This would enable the conclusion to concentrate more on the main findings of the study, the acknowledged limitations, and the directions for future research.

Thanks for your comment, we have revised that as follows:

This study investigated bank risk assessment in the context of international trade settlement using a hybrid FNN optimized via a GA, benchmarking it against conventional statistical and machine learning models. Beyond methodological comparison, our analysis focused on how different datasets affect model performance and interpretability. The datasets used—Bank Marketing [27], Lending Club [28], and the German Credit Risk dataset—differ significantly in data structure, feature richness, and class distribution, which directly influenced the model’s behavior. For example, the German Credit dataset has a limited number of features and is relatively balanced, allowing the FNN model to converge quickly with high accuracy. In contrast, the Lending Club dataset contains high-dimensional and imbalanced loan data, which challenged general classifiers such as LR and SVM, yet highlighted the strength of FNN in handling fuzzy boundaries between risk categories. Variable influence analysis revealed that features such as “loan purpose,” “employment length,” and “default history” had varying degrees of impact on the final classification depending on the dataset. In the Lending Club dataset, borrower intent and historical performance played a dominant role, whereas in the German dataset, credit history and savings level were more influential. The inclusion of fuzzy membership functions allowed FNN to capture these nonlinear, context-dependent relationships more effectively than crisp models.

Regarding model optimization, a comparative test between GA, PSO, and SA showed that while PSO exhibited faster convergence in early iterations, GA achieved more stable and higher-quality solutions over repeated runs. Despite GA's conceptual simplicity, its performance in refining rule weights and membership parameters within the FNN framework proved effective for our credit risk classification task. While our evaluation primarily relied on accuracy, this decision was driven by class balance in the datasets and the goal of evaluating holistic model performance. Nonetheless, we acknowledge that incorporating additional metrics such as F1-score and sensitivity would provide more nuanced insight—especially in datasets with class imbalance like Lending Club. Future research should expand the indicator set to include macroeconomic variables and behavioral features, which could further improve prediction under volatile market conditions.

Response to Reviewer 2:

1. Scientific Methods: The manuscript applies Genetic Algorithm (GA) and Fuzzy Neural Networks (FNN) for risk assessment in international trade settlements. Both methods are appropriate for handling the non-linearity, uncertainty, and optimization tasks involved. The method description is reasonably detailed. However, I think there needs to be a clearer explanation of the pros and cons of each method to compare with the application of the GA-FNN method.

The integration of GA with FNN aims to exploit the strengths of both methods by using GA to optimize FNN parameters such as membership functions, fuzzy rule weights, and network topology. The specific optimization objective is to minimize classification error on a credit risk dataset while maximizing generalization performance on unseen data. This hybrid model addresses both the structural learning of the FNN and its parameter tuning, ensuring that the resulting model is not only accurate but also adaptable to varying data distributions. In the context of bank risk analysis, this means more robust prediction of borrower default risk under diverse economic conditions. This section thus serves as a critical link between algorithmic design and its applied relevance, bridging the gap between methodological development and financial practice.

2. Experimental Design: Three well-known datasets (Bank Marketing Dataset, Lending Club, and German Credit Risk) were used. The authors performed comparative experiments against baseline models (RAROC, LR, BPNN, SVM, FNN) and alternative optimizations (SA, PSO). This structure is good and strengthens the credibility of the claims.

Thanks for your comment.

3. Controls and Comparisons: Proper comparisons with traditional methods and alternative optimization methods are included. However, some baseline models (RAROC, LR) could have been described more thoroughly in terms of parameter tuning.

For the RAROC model, expected loss (EL), economic capital (EC), and cost of capital were calculated using standard Basel formulas, with EL = PD × LGD × EAD, and E

---

## [Decision Letter · Decision Letter 1]

Design of an evolutionary model for international trade settlement based on genetic algorithm and fuzzy neural network

PONE-D-25-18703R1

Dear Dr. Tu,

We’re pleased to inform you that your manuscript has been judged scientifically suitable for publication and will be formally accepted for publication once it meets all outstanding technical requirements.

Kind regards,

Ardashir Mohammadzadeh, Phd

Academic Editor

PLOS ONE

Additional Editor Comments (optional):

Reviewers' comments:

Reviewer's Responses to Questions

**Comments to the Author**

Reviewer #1: All comments have been addressed

Reviewer #3: All comments have been addressed

2. Is the manuscript technically sound, and do the data support the conclusions?

Reviewer #1: Yes

Reviewer #3: Yes

3. Has the statistical analysis been performed appropriately and rigorously?

Reviewer #1: No

Reviewer #3: Yes

4. Have the authors made all data underlying the findings in their manuscript fully available?

Reviewer #1: Yes

Reviewer #3: Yes

5. Is the manuscript presented in an intelligible fashion and written in standard English?

Reviewer #1: Yes

Reviewer #3: Yes

Reviewer #1: I am pleased that the authors have now fully addressed the comments made in my last review, and the resulting paper is better structured and cleaner for the reader.

I would still encourage the authors to revise one specific aspect in Section 4.1 (Experiment Setup). The authors have fulfilled my previous request by adding particular details about the dataset split. The manuscript shows that the data was divided into training (70%), validation (15%), and test (15%) sets, but the rationale for this choice is not explained. It also remains unclear whether the split was random or stratified by class and whether a fixed random seed was used. I recommend clarifying the splitting strategy and indicating the seed, if applicable.

Reviewer #3: As one of the first evaluators, and including the other two reviewers' comments, the authors reflect on the questions raised from all and for the comments, they revised kindly, for me its good enough to publishing in this journal

**Do you want your identity to be public for this peer review?** For information about this choice, including consent withdrawal, please see our Privacy Policy

Reviewer #1: No

Reviewer #3: **Yes: ** Zewdu Eskezia Gelaye(Research Assistant Professor of Accounting and Finance)

---

## [Editor Report · Acceptance letter]

PONE-D-25-18703R1

PLOS ONE

Dear Dr. Tu,

I'm pleased to inform you that your manuscript has been deemed suitable for publication in PLOS ONE. Congratulations! Your manuscript is now being handed over to our production team.

Kind regards,

on behalf of

Dr. Ardashir Mohammadzadeh

Academic Editor

PLOS ONE